# Future of Endoscopic Spine Surgery: Insights from Cutting-Edge Technology in the Industrial Field

**DOI:** 10.3390/bioengineering10121363

**Published:** 2023-11-27

**Authors:** Woon-Tak Yuh, You-Sang Lee, Jong-Hyeok Jeon, Il Choi

**Affiliations:** 1Department of Neurosurgery, Hallym University Dongtan Sacred Heart Hospital, 7 Keunjaebong-gil, Hwaseong-si 18450, Gyeonggi-do, Republic of Korea; woontak.yuh@gmail.com; 2Department of Neurosurgery, Hallym University College of Medicine, 1 Hallym Daehak-gil, Chuncheon-si 24252, Gangwon-do, Republic of Korea; 3R&D Team, Solendos Inc., 503 38-21 Digital-ro 31-gil Guro-gu, Seoul 08376, Republic of Korea; yslee@solendos.com; 4Biounit Co., Ltd., 810~817, WB113, Misagangbyeonjungang-ro, Hanam-si 12939, Gyeonggi-do, Republic of Korea; jhjeon3459@biounuit.co.kr

**Keywords:** endoscope, endoscopic spinal surgery, surgical techniques, camera, visualization technology, radiofrequency ablation, drill

## Abstract

In the evolving landscape of spinal surgery, technological advancements play a pivotal role in enhancing surgical outcomes and patient experiences. This paper delves into the cutting-edge technologies underpinning endoscopic spine surgery (ESS), specifically highlighting the innovations in scope cameras, RF equipment, and drills. The modern scope camera, with its capability for high-resolution imaging, offers surgeons unparalleled visualization, enabling precise interventions. Radiofrequency (RF) equipment has emerged as a crucial tool, providing efficient energy delivery for tissue modulation without significant collateral damage. Drills, with their enhanced torque and adaptability, allow for meticulous bone work, ensuring structural integrity. As minimally invasive spine surgery (MISS) becomes the standard, the integration and optimization of these technologies are paramount. This review captures the current state of these tools and anticipates their continued evolution, setting the stage for the next frontier in spinal surgery.

## 1. Introduction

Endoscopic Spinal Surgery (ESS) stands as a significant leap in the direction of minimally invasive spinal surgery (MISS). It offers an avenue to enhance patient outcomes while reducing procedural invasiveness [1,2,3,4,5,6,7]. Unlike established endoscopic procedures such as laparoscopic, arthroscopic, retroperitoneal, or gastrointestinal (GI) endoscopic surgeries and procedures—which benefit from the natural anatomical working spaces of peritoneal cavities, joint cavities, and the lumen of GI organs—spinal endoscopic surgery faced the unique challenge of lacking an inherent anatomical working space. This limitation initially posed a hurdle to the onset of ESS. It was first overcome by leveraging Kambin’s triangle as a surgical corridor to access the epidural space in lumbar disc surgeries [8,9].

The evolution of ESS began with the emergence and development of Full-Endoscopic Spinal Surgery (FESS), subsequently followed by Unilateral Biportal Endoscopic Spine Surgery (UBE) [10,11]. While both surgical techniques offer the benefits of minimally invasive procedures with excellent visualization and are widely used, they exhibit distinct differences primarily stemming from the design of their devices. In FESS, the endoscope serves roles as both visualization and providing a channel for instruments. Conversely, UBE utilizes the endoscope solely for visualization. This difference brings about specific advantages and disadvantages for each technique, broadening the range of procedures and catering to diverse surgical scenarios and preferences.

The development of surgical devices has played a pivotal role in advancing ESS [12]. The advent of modern scope cameras equipped with high-resolution imaging and cutting-edge sensor technology has significantly enhanced visualization, thereby facilitating more precise interventions. Similarly, innovations in Radiofrequency (RF) equipment have enabled safe bleeding control with minimal collateral damage. The refinement in drill technology, characterized by superior torque, speed, and adaptability in water-based surgery, has been instrumental in performing powerful and delicate bone drilling procedures while preserving structural integrity.

Endoscopic Spine Surgery (ESS) has undergone a remarkable evolution, beginning with the introduction of transforaminal (trans-Kambin) lumbar discectomy by Parviz Kambin in the 1980s [8], followed by endoscopic interlaminar lumbar discectomy introduced by Ruetten et al. in the 2000s [13]. Over the decades, the indications for ESS have expanded significantly, now encompassing surgeries from cervical to lumbar segments and evolving to include not only decompressive but also fusion surgeries. In 2020, the AO Spine nomenclature system was established [14], categorizing ESS types based on approach corridor, visualization, spinal segments, and specific procedures. This comprehensive system includes various ESS techniques such as Anterior Endoscopic Cervical Discectomy (AECD) [15], Posterior Endoscopic Cervical Foraminotomy (PECF), Posterior Endoscopic Cervical Discectomy (PECD) [16], Cervical Endoscopic Unilateral Laminotomy for Bilateral Decompression (CE-ULBD) [17], Transforaminal Endoscopic Thoracic Discectomy (TETD) [18], Thoracic Endoscopic Unilateral Laminotomy for Bilateral Decompression (TE-ULBD) [19], Transforaminal Endoscopic Lumbar Discectomy (TELD) [20], Interlaminar Endoscopic Lumbar Discectomy (IELD) [21], Extraforaminal Endoscopic Lumbar Discectomy (EELD) [22], and Transforaminal Endoscopic Lumbar Foraminotomy (TELF) [23]. Additionally, techniques like Transforaminal Endoscopic Lumbar Interbody Fusion [24] and UBE Fusion [25,26] are also included. Beyond the AO Spine nomenclature, ESS is also employed in various surgeries, such as surgery for craniovertebral junction [27], spinal metastasis [28], spinal abscess [29], and far-out syndrome [30]. The rapid adoption and surging popularity of ESS underscore its capacity to effectively address a wide range of spinal conditions, covering almost all segments and pathologies in spine surgeries. However, given that current ESS systems are tailored to water-based surgery, it is challenging to perform surgeries for intradural pathologies, such as intradural tumors, arachnoid cysts, and hematomas since the infused saline could lead to increased intracranial pressure (IICP) during the operation [31].

As the realm of MISS expands, the ongoing integration and enhancement of current technologies and techniques remain imperative. Despite the progress in ESS technologies and surgical techniques, a chasm still exists between the present capabilities and the ideal goals that endoscopic spine surgeons aim to achieve.

Looking ahead, the field is ready for more innovation. By assimilating concepts and technologies from other endoscopic surgical disciplines, evolving device technologies, and incorporating advanced tools like Artificial Intelligence (AI) and Augmented Reality (AR), the horizon seems promising. This review aims to provide a thorough examination of the current state of technologies and techniques in ESS while looking forward to the exciting future possibilities in this field.

## 2. Surgical Techniques of Endoscopic Spinal Surgery

With decades of advancement, both FESS and UBE have established themselves as premier minimally invasive surgical techniques in spine surgery. Each technique offers unique advantages and faces distinct challenges. The primary differentiation between the two lies in the endoscopic scope’s role. In FESS, the endoscope serves dual purposes: visualization and providing a channel for instruments [2,3,32]. Conversely, UBE utilizes the endoscope solely for visualization. This means that in FESS, the visualization and instrument angles align, while in UBE, there is typically a 30-degree angular difference. Such variations significantly influence surgical tactics (Figure 1).

FESS’s surgical pathway can be likened to a rifle’s barrel: both the camera systems and surgical instruments pass through a single, slender, and elongated rigid scope. This design, tailored for accessing narrower spaces like the transforaminal corridor, makes FESS suitable for keyhole surgeries requiring minimal bony drilling. Nevertheless, it necessitates the use of extended and slim instruments, posing challenges to crafting versatile, robust surgical tools. Here, the scope’s diameter plays a critical role and presents a trade-off. Earlier FESS models, with an outer diameter of approximately 7 mm and an inner channel of 4 mm, were apt for the foraminal area but restricted instrument variety. However, contemporary designs, boasting up to 11 mm outer and 4.6 mm inner diameters, accommodate a wider range of instruments, with the inclusion of larger drill burrs aiding efficient bone work.

Conversely, UBE’s design is familiar to many spine surgeons due to its resemblance to traditional open surgeries, particularly the 30-degree angle between the scope and the working channel. The shorter working portal in UBE accommodates most traditional open surgery instruments, including large-sized high-speed drills, various Kerrison rongeurs, curettes, and dissectors. However, the angular difference between the scope and working instruments in UBE necessitates more extensive bone work than FESS, making access to the actual working spaces through the drilled-out bone more challenging. This inherent trade-off makes UBE less optimal for minimal keyhole approaches.

## 3. Advancements in Endoscopic Surgery Devices

### 3.1. Camera and Lighting Systems

#### 3.1.1. Resolution

Resolution refers to the number of pixels a display holds. For instance, 2K (Full HD) has a resolution of 1920 × 1080 pixels, whereas 4K (Ultra HD) boasts a resolution of 3840 × 2160 pixels. The advantage of higher resolutions is their capacity to depict patient tissues with greater precision. Hospital monitors are usually fixed in size, predominantly ranging between 27 and 32 inches. Displaying a 4K resolution on the same monitor size as 2K essentially packs in more pixels, resulting in enhanced detail.

Recent advances in camera lens technology have unveiled previously hidden microstructures, such as the epidural ligaments. Recognizing these structures has refined dissection techniques and improved bleeding control, thus enhancing surgical safety. The evolution from analog Standard Definition (SD) to digital formats—encompassing High Definition (HD), Full HD, Quad HD, and Ultra HD—stretches from the scope of capturing surgical scenes to the screen that showcases them. These innovations herald a future where even intricate anatomical structures are distinctly visible, especially under magnification.

#### 3.1.2. Color and Camera Sensor

The authenticity of color representation in surgical imaging is paramount, directly impacting the ability of surgeons to distinguish between different tissues and discern minute changes that may have clinical significance. Color representation in surgical endoscopy is greatly influenced by the camera sensor integrated within the camera head.

Historically, CCD (Charge-Coupled Device) sensors were the gold standard, chosen for their superior color fidelity, even though they typically had a lower resolution and came with higher production costs. Over the past decade, however, there has been a notable shift towards CMOS (Complementary Metal–Oxide-Semiconductor) sensors across most manufacturers. CMOS sensors, being more cost-effective and boasting higher resolution, have gained widespread popularity. However, they do tend to slightly compromise on color quality when compared to CCD sensors, as outlined in Table 1.

Further differentiation in camera sensors comes in the form of one-chip or three-chip systems. White light comprises red, blue, and green colors. A three-chip system has individual sensors for each of these colors, ensuring richer and more accurate color representation. On the other hand, a one-chip system captures all three primary colors with a single sensor, effectively reducing the color data by a third (Figure 2). The choice between a one-chip and a three-chip system is largely a matter of market positioning. While three-chip systems are tailored for high-end markets due to their superior imaging capabilities, one-chip systems are designed with cost-effectiveness in mind, making them more suited for low-end markets [34].

#### 3.1.3. Lighting System

The evolution of lighting systems has paralleled the advancements in light technology. Transitioning from the erstwhile Xenon and Halogen lamps, the predilection now is towards LED lamps. However, the method of light delivery has retained its traditional approach of harnessing a separate light source and channeling it via fiber optics. Similar to various endoscopic surgeries in other fields.

#### 3.1.4. Angles of Scope and Flexibility

FESS started with a rigid scope for both visualization and the working portal, while UBE, drawing from laparoscopic surgery, uses a short rigid scope with a separate working portal. Commonly used scope angles are 0-degree and 30-degree, but 45-degree scopes are also employed to visualize all corners of the working spaces. Currently, there is no commercially available semi-rigid scope with an adjustable terminal angle, like in da Vinci robotic surgery, or a flexible scope as in GI endoscopy.

### 3.2. Radiofrequency Ablation Systems

In endoscopic surgeries, bleeding is a significant issue as it obscures the surgical field, potentially leading to delays or even cessation of the procedure. It is imperative to accurately and robustly control bleeding without causing damage to neural tissues. The extent of heating damage can vary depending on the RF (Radiofrequency) output and power, affecting uncontrolled areas. Hence, the development of tools that can precisely control the range and area of effect is crucial in surgeries.

Since the advent of spinal endoscopic surgery, there has been significant progress in RF technology, alongside a substantial accumulation of data from actual surgeries. Initially, its application was limited to a certain extent, primarily used on soft tissues. However, as the technology advanced, its application expanded not only for soft tissue but also for bone bleeding control. Furthermore, RF functionality has evolved from being merely used for coagulation to tissue ablation as well [35]. The evolution of Electrosurgical Unit (ESU) technology can be broadly categorized into the advancement of generators and electrodes.

#### 3.2.1. Generator

Typically, Radiofrequency denotes a frequency band of 3 kHz to 300 GHz, among which the frequency band mainly used in ESU ranges from 100 kHz to 5 MHz. The properties of electromagnetic radiation dictate that energy (E) is directly proportional to frequency (μ) and inversely proportional to wavelength (λ). Hence, as the frequency increases, the wavelength shortens, and higher energy is generated. Generators operating in MHz units tend to use lower output levels (Voltage or Watt) than those operating in kHz units due to the high energy generated [36]. Despite reaching the desired target temperature quickly, MHz generators have their output range limited to a lower range, making fine-tuning the output stages somewhat challenging, as shown in Figure 3.

Moreover, human tissues have unique impedance values, meaning that even with the same output from the generator, the resulting output varies depending on the tissue type [37]. Therefore, there is a growing demand for the development of generators capable of more precise output adjustments to achieve appropriate coagulation and ablation across different tissues.

#### 3.2.2. Electrode

Electrodes are categorized as either Monopolar or Bipolar based on their polarity [38,39]. Monopolar devices consist of an active electrode at the tip and a return electrode on a pad, and they are commonly used in general surgeries. However, if the return electrode (or pad) is not correctly placed, it can cause current concentration, leading to complications like burns [40]. On the other hand, Bipolar devices have both active and return electrodes at the tip, significantly reducing the risk of complications like burns in the pad area (Figure 4a).

A notable example of a Bipolar electrode is the Bovie type. Traditional Bovie electrodes have separate shafts for each pole, making them unsuitable for surgeries in tight spaces like endoscopic operations. To address this limitation, the Wand-type bipolar electrode was developed for endoscopic surgeries (Figure 4b). This design has both active and return electrodes on a single shaft, making it usable even in narrow-diameter endoscopic scopes. The advent of such Wand-type bipolar electrodes has enabled the application of ESUs even in smaller-diameter spinal endoscopic surgeries. For ablation procedures, a higher current output than coagulation is applied. As surgical techniques advance, it’s anticipated that future electrode developments will emphasize miniaturized, mechanically and electrically durable designs capable of enhanced ablation.

### 3.3. Drill Systems

In endoscopic surgeries, drills play a vital role in decompressing neural tissues. However, in spinal endoscopic surgery, drills require several unique characteristics differing from other endoscopic surgeries, such as transsphenoidal approach surgery.

Firstly, spinal endoscopic surgeries are conducted in a water-based environment, necessitating certain features for drill operation underwater. It is crucial to swiftly expel bone dust generated during drilling to maintain a clear surgical field. To achieve this, a smooth out-flow maintained through water flow is essential. For instance, in FESS, Wolf’s drill has a suction attached to improve water flow artificially. In UBE surgery, maintaining a clear field of view becomes challenging without a separate suction for out-flow, especially if the drill’s burr size is too large or the shaft is too thick, occupying part of the out-flow pathway, which deteriorates visibility. Hence, endoscopic drills need to be specialized towards having a thin shaft capable of accommodating large burrs and exhibiting high stability at high speeds.

Secondly, given the water-based surgery nature, waterproof capabilities, especially in the handpiece, are indispensable for drill durability. However, the current drills have been developed without much consideration for this aspect. Frequent malfunctions, which could lead to surgery interruption, pose significant economic burdens on the operating hospitals, making it challenging to actively perform endoscopic surgeries. Industrial products have a step-by-step certification for waterproof functionality, as indicated by the IP (Identification code for Protection) ratings, yet such certification has not been applied to spinal drills, signifying a need for such certification in endoscopic surgery equipment. This issue has been attempted to be mitigated in clinical settings through various methods, such as attaching waterproof features in-between to enhance durability.

Lastly, the current drills used in spinal endoscopic surgery have been adapted from the shaver types used in arthroscopic surgeries for shoulders and knees and high-speed drills used in conventional spine surgery. The shaver type offers better torque and the possibility to attach suction, maintaining a clear surgical field, albeit at the cost of being heavier and having a slower drill head speed. On the other hand, high-speed drills are lighter with the advantage of applying high RPM (Revolutions Per Minute) methods but suffer from weaker torque holding the drill, posing durability issues. Given the unique condition in spinal endoscopy where lesions are far from the skin, requiring bone resection at distant locations, there is a demand for the development of specialized drills exclusively for spinal endoscopic surgery.

## 4. Discussion and Future Directions

### 4.1. Development of ESS

ESS evolved for decades from FESS to UBE and is currently becoming a mainstream of MIS spine surgery. Devices and technologies concurrently have developed, adopting concepts of prior endoscopic surgery from other fields, such as laparoscopic surgery and arthroscopic surgery. Thus, basic concepts are similar, such as a rigid scope and straight surgical instruments. However, there are other successful endoscopic surgery fields, such as retroperitoneal robotic surgery and gastrointestinal endoscopic procedures. They have distinct differences in terms of surgical techniques, and there could be advances that are able to be adopted in the case of ESS.

One of the main limitations of current ESS techniques is the use of single working instruments with minimal assistance in terms of a scope retractor, which makes fine tasks, such as intramedullary surgery or dura suture, challenging. In comparison, traditional three-portal laparoscopic surgery with one camera portal and two working portals allows the surgeon to operate with both hands. This approach provides a more dynamic and ergonomic way to manipulate tissues and organs, potentially reducing surgeon fatigue and improving precision. Robotic surgery systems, such as the da Vinci Surgical System, represent a further advancement, with multiple robotic arms providing the surgeon with the capability to manipulate several instruments concurrently. The enhanced dexterity and precision of robotic arms, along with three-dimensional visualization, could allow for complex maneuvers that are difficult to perform with manual endoscopic instruments. The application of such laparoscopic or robotic technology to ESS could revolutionize the field by enabling more intricate and delicate spine surgeries with potentially improved outcomes and lower rates of complications.

Another issue to consider with ESS is its reliance on water-based surgery. Since water must be constantly infused during the procedure in the current ESS system, there is a risk of increased intracranial pressure (IICP) if an incidental durotomy occurs. This risk limits the indications for ESS to extradural surgery. To overcome this limitation and expand the indications of ESS to include intradural pathology, such as intradural tumors, it may be necessary to switch to air-based surgery after performing the dural incision. Additionally, strategies must be in place to manage minor bleeding from surrounding structures.

### 4.2. Camera and Visualization

ESS faces complex and narrow anatomical structures in many cases. For example, it is difficult to visualize the thecal sac, traversing nerve roots, and exiting roots in foraminal and extraforaminal areas at once due to the narrow bony corridor. To overcome this, angled scopes are widely used, but the angle is fixed, and visualization is often not enough. The da Vinci robotic surgery system provides a freely movable distal camera. The GI endoscope is much more flexible when passing the complex curves of the GI organ lumen; however, the working channel is limited. If we can adopt this flexible or semi-rigid camera concept, ESS could be enhanced one step further.

During endoscopic surgery, securing a broader field of view can enhance surgical safety. Several candidate technologies can be considered to achieve this wider view. These include using a wide-angle lens, mounting multiple lenses in various directions on a single scope, or employing a flexible scope that allows the tip to rotate, providing a more expansive surgical field of view. However, current technological limitations exist. Using a wide-angle lens can cause distortion at the periphery of the view, and incorporating multiple lenses might result in a thicker scope. Nevertheless, the fact that other clinical fields are already utilizing scopes produced by Olympus, where the tip can flex, suggests that the day when such technologies are applied to spinal endoscopic surgeries to ensure a broader view might not be far off.

### 4.3. Wireless System

For scopes connected to the system, in the case of a full endoscope, there are three lines: the water line, the suction line, and the image line. In bi-portal surgery, two lines are connected: the water line and the imaging line. As the number of connected lines increases, the weight of the scope becomes heavier, imposing limitations on its movement. Therefore, efforts should be made to reduce the number of lines. Technologically, to eliminate the currently used imaging line, one can consider a method where the image data obtained from the scope are converted in a device and then wirelessly transmitted after an image conversion process. The console then receives these data and converts the data back into an image. If this entire process can be completed in less than 130 ms [41], surgeons will not experience any discomfort during surgery. However, if it exceeds 130 ms, they might feel inconvenienced due to the delay. At present, using wireless technology might make the scope heavier and compromise image quality. However, as technology advances, these challenges are likely to be overcome [41] (Figure 5).

### 4.4. Coagulation and Ablation Tools

The ideal coagulation and ablation tool should have the capability to control bleeding adequately and precisely ablate the targeted tissue. This entails the challenging juxtaposition of having sufficient power and the coagulation of selective tissues. From the early stages of endoscopic spine surgery, lasers have been applied for tissue ablation. While their efficacy is commendable, there have been limitations to their widespread use due to issues like damage to normal tissues from non-selective tissue removal, the high cost of generators, and noise. We look forward to the development of tools that can remove tissue more effectively and selectively.

### 4.5. Drill System

#### 4.5.1. Waterproofing and Durability

Endoscopic surgery, being water-based, inherently has durability as an issue. Water refluxed from the surgical field can enter the electric drill motor and bearing system, causing them to rust and disrupting the transmission of electricity. To address this, research is being conducted on methods like connecting the drill tip to the handpiece using a waterproof block that has a waterproofing effect. For surgeons to perform surgery without concerns about instrument damage during the operation, enhancing the waterproofing function is an essential aspect of the advancement of endoscopic spine surgery.

#### 4.5.2. High Speed and High Torque Even with Long Thin Shaft

The RPM and torque of the spinal endoscopic drill adhere to the formula RPM/torque = constant. The constant value in this equation has a proportional relationship with the given electrical power. Moreover, the further the drill tip is from the drill attachment, the weaker the torque power becomes. Therefore, with the advancement of technology, an increase in the constant value is expected to promote the development of high RPM drills with stronger torque power.

## 5. Conclusions

Endoscopic spinal surgery, when compared to endoscopic surgeries in other parts of the body, presents unique challenges: the absence of an open space or pathway, the lesion being distant from the skin, and the need for extensive bone work. Additionally, the requirement to preserve neural tissues has been a critical aspect, underscoring the importance of advancements in endoscopic surgical techniques. A discernible gap exists between the evolution of industrial technology and the clinical needs in the field. Through this review paper, we have examined this gap. We anticipate that the emergence of new technologies will bridge this divide, paving the way for further advancements in the domain of endoscopic spinal surgery.

## Figures and Tables

**Figure 1 bioengineering-10-01363-f001:**
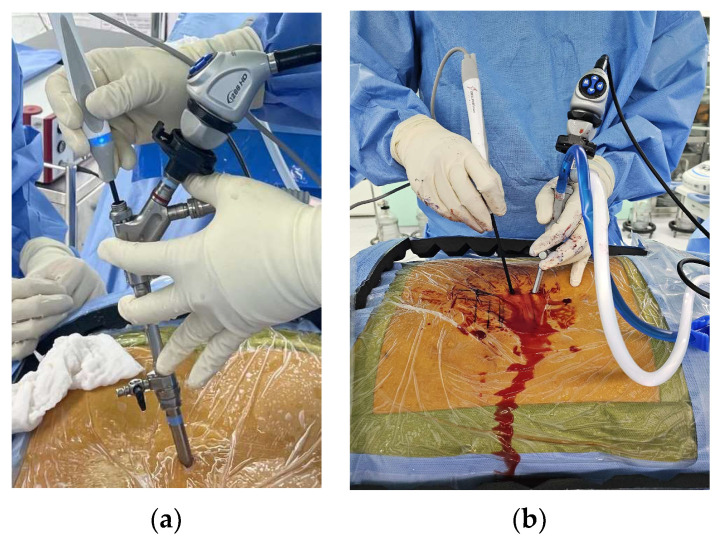
Device designs of Full-Endoscopic Spinal Surgery (FESS) and Unilateral Biportal Endoscopic Spine Surgery (UBE): (**a**) FESS aligning scope and working instruments; (**b**) UBE with 30-degree angulation between scope and working instruments.

**Figure 2 bioengineering-10-01363-f002:**
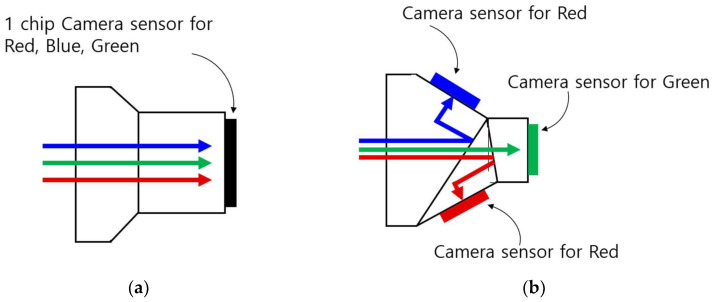
Different chip systems in camera sensors: (**a**) single-chip system; (**b**) three-chip system.

**Figure 3 bioengineering-10-01363-f003:**
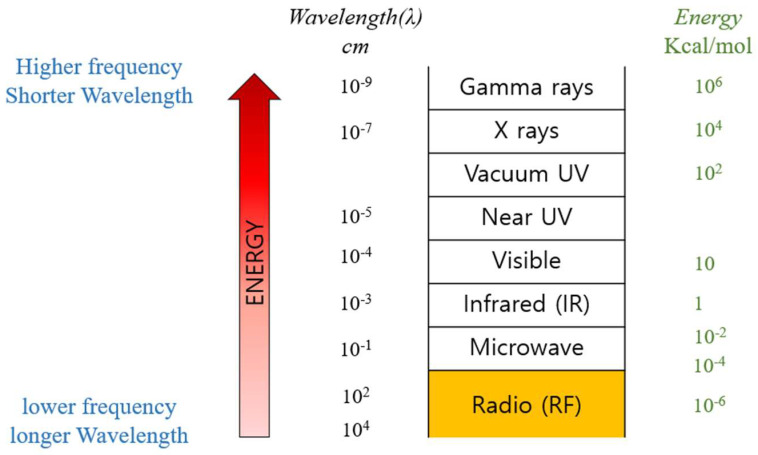
Energy of each type of electromagnetic radiation.

**Figure 4 bioengineering-10-01363-f004:**
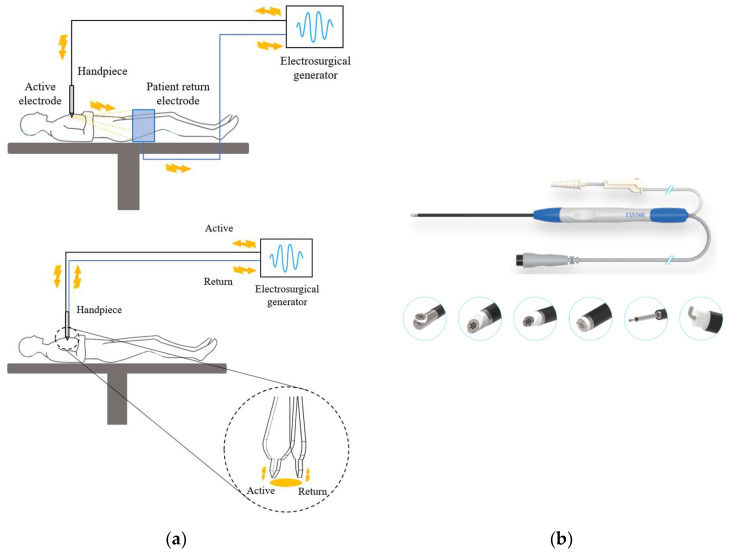
Different types of radiofrequency ablation systems: (**a**) monopolar and bipolar configuration of the circuit; (**b**) Wand-type bipolar electrodes.

**Figure 5 bioengineering-10-01363-f005:**
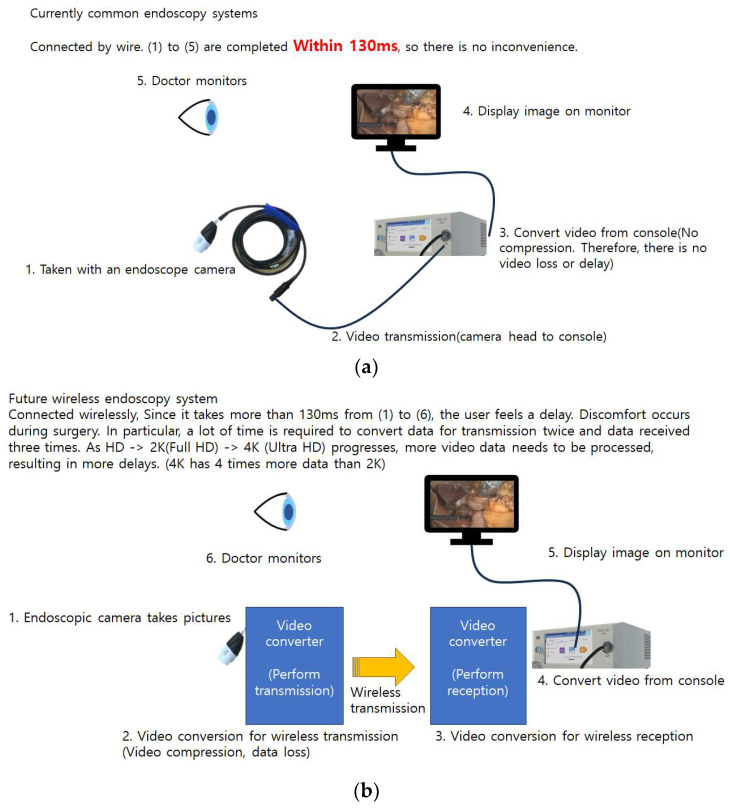
Video Transmission in Current Wired vs. Future Wireless Endoscopy Systems. (**a**) A current wired system that achieves video transmission to the monitor in under 130ms, ensuring no noticeable delay. (**b**) A future wireless system transmitting wirelessly. At present, this method faces challenges of transmission delays exceeding 130ms, particularly with higher-resolution 4K video, due to the data processing required for wireless transmission.

**Table 1 bioengineering-10-01363-t001:** Comparison of CCD and CMOS sensors of color quality, resolution, and cost [33].

	CCD	CMOS
Image Quality	Excellent color representation Less light saturation Clear view in dark area	Compromised color representation More light saturation Dark areas less visible
Production Cost	10 times higher than CMOS	Economical
Resolution	Maxes out at Full HD (2K)	Progressing from 2K to 8K

## Data Availability

Data sharing is not applicable to this review.

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
