# Peer review of "Future of Endoscopic Spine Surgery: Insights from Cutting-Edge Technology in the Industrial Field"

_bioengineering, 2023, doi:10.3390/bioengineering10121363_

Round 1

Reviewer 1 Report

Comments and Suggestions for Authors

This paper delves into the cutting edge technologies underpinning endoscopic spine surgery, specifically highlighting the innovations in scope cameras, RF equipment, and drills.

This review captures the current state of these tools and anticipates their continued evolution, setting the stage for the next frontier in spinal surgery. 

Comments on the Quality of English Language

No

Reviewer 2 Report

Comments and Suggestions for Authors

In this review, the authors aim to delve into endoscopic spine surgery (ESS), specifically highlighting the innovations in scope cameras, radiofrequency (RF) equipment, and drills. As minimally invasive spine surgery (MISS) becomes the standard, the integration and optimization of these technologies are paramount. This review not only captures the current state of these tools but also envisions their future, setting the stage for the next frontier in spinal surgery. Based on my evaluation, I believe that this work has the potential to make a significant contribution to the field. To make the manuscript more profound, we suggest adding the following modifications:

 1.     Please add "radiofrequency" before the abbreviation "RF" in the line 16 at the abstract.

2.     In the introduction, it would be beneficial to discuss the types of spinal surgeries that commonly employ ESS techniques.

3.     In the Discussion and Future Directions section, it would be valuable to mention the companies that produce ESS devices and their role in advancing this field.

Reviewer 3 Report

Comments and Suggestions for Authors

Congratulations on a remarkable paper that tackles such an important aspect of spinal surgery. Your comprehensive review of the technologies that are shaping endoscopic spine surgery is commendable. The details regarding high-resolution imaging from modern scope cameras, efficient energy delivery via RF equipment, and the precision of current drills are particularly noteworthy. Your work not only provides a current snapshot of these technological advancements but also offers a visionary outlook on the future integration of AI and real-time feedback mechanisms. Your contribution to minimally invasive spine surgery is significant, and I applaud your efforts in advancing our understanding of this evolving field.
